# Global stroke burden attributable to household air pollution: Insights from GBD 2021 and projections to 2040

Manqing Li[1,2], Shicheng Liao[2], Chao Wang[2], Haojia Ma[2], Xiumei Xing[1], Jianhui Yuan[3], Jie Jiang[2]*, Zhuoying Zeng[2]*

1 School of Public Health, Sun Yat-sen University, Guangzhou, China, 2 Chemical Analysis & Physical Testing Institute, Shenzhen Center for Disease Control and Prevention, Shenzhen, China, 3 Nanshan District Center for Disease Control and Prevention, Shenzhen, China

* zengzhuoying@szu.edu.cn (ZZ); 1071455735@qq.com (JJ)

## Abstract

### Objectives

To analyze the global stroke burden attributable to household air pollution (HAP) using Global Burden of Disease (GBD) 2021 data, examine its spatiotemporal patterns from 1990–2021, and project future trends through 2040.

### Study design

Retrospective analysis using GBD data with future projections

### Methods

Analysis of age-standardized rates, deaths, disability-adjusted life years (DALYs), years lived with disability (YLDs), and years of life lost (YLLs) due to HAP-related stroke across 204 countries. The linear regression model examined global time trends. Cluster analysis investigated the patterns of disease burden changes across GBD regions. The Nordpred model projected trends up to 2040.

### Results

In 2021, HAP caused 1.23 million stroke deaths and 26.78 million DALYs globally. From 1990–2021, age-standardized death rates (EAPC = −0.37), DALY rates (EAPC = −0.20), and YLL rates (EAPC = −0.30) decreased, while YLD rates increased (EAPC = 1.11). Males showed a higher burden than females, with peak rates in the 80–84 age group. Middle-SDI regions had the highest age-standardized rates, with East Asia recording the largest absolute numbers. South and East Asia showed significant increases, while Western Europe, Eastern Europe, and High-income North America showed the greatest declines. Predictions using the Nordpred model indicated rising absolute numbers of deaths (to 1.79 million), DALYs (to 34.76 million),

**Data availability statement:** The data underlying the results presented in the study are publicly available from the Global Burden of Disease (GBD) database. The GBD database can be accessed at https://ghdx.healthdata.org/.

**Funding:** 1. Shenzhen Key Medical Discipline, with grant number SZXK066, received by Jie Jiang. 2. Shenzhen Postdoctoral Research Fund for Staying in Shenzhen, received by Zhuoying Zeng. 3. Guangdong Basic and Applied Basic Research Foundation, with grant number 2022A151511119, received by Zhuoying Zeng. 4. Shenzhen Science and Technology Program, with grant number JCY20220530150402004, received by Zhuoying Zeng. The funders had no role in study design, data collection and analysis, decision to publish.

**Competing interests:** The authors have declared that no competing interests exist.

YLDs (to 3.29 million), and YLLs (to 31.39 million), with males consistently bearing a higher burden, though ASRs are expected to decline for both sexes.

## Conclusions

Our findings suggest that regions with higher economic development and greater adoption of clean energy are associated with lower HAP-related stroke burden, possibly through improvements in indoor air quality. The observed regional and gender disparities emphasize the need for targeted interventions, particularly in less developed regions and among high-risk groups such as men and older adults.

## 1. Introduction

Household air pollution (HAP) is a significant global health challenge, especially in low- and middle-income countries where traditional fuels such as wood, coal, and biomass are widely used for cooking and heating. The combustion of these fuels generates harmful pollutants, including particulate matter (PM), carbon monoxide (CO), sulfur dioxide ($SO_2$), and volatile organic compounds (VOCs), which severely compromise indoor air quality and pose substantial health risks [1–3]. According to the World Health Organization (WHO), HAP accounts for approximately 3.1 million premature deaths annually, with the burden disproportionately borne by developing countries [4,5]. Among its numerous adverse health effects, HAP has been strongly associated with respiratory and cardiovascular diseases [6–9], both of which are epidemiologically linked to the pathophysiology of stroke.

Stroke remains one of the leading causes of death and disability worldwide, resulting from a sudden interruption of blood supply to the brain that causes brain tissue damage [10,11]. Globally, stroke accounts for over 7 million deaths annually and leaves millions more with long-term disabilities, significantly impairing quality of life and imposing substantial healthcare costs [11,12]. While traditional risk factors for stroke, such as hypertension, diabetes, and smoking, are well-documented, growing evidence highlights HAP as an emerging risk factor [13]. Exposure to pollutants like PM2.5 can induce chronic inflammation, oxidative stress, and endothelial dysfunction, mechanisms that contribute to the development of atherosclerosis and stroke. Despite the increasing recognition of the link between HAP and stroke, the specific global burden of stroke attributable to HAP remains poorly characterized. Previous analyses have often focused on single countries or regions [14,15], and global-scale studies are limited and outdated. For instance, Lu et al. [16] provided a global assessment of HAP-related stroke burden using Global Burden of Disease (GBD) 2019 data, estimating 0.6 million stroke deaths and 14.7 million DALYs in 2019, but their analysis lacked the latest data, detailed gender and age stratifications, and long-term projections. Furthermore, the spatiotemporal trends of HAP-related stroke burden over recent decades and projections for future trends have not been systematically explored. Understanding these patterns is crucial for identifying high-risk populations and regions and for informing targeted public health interventions.

To address these gaps, this study utilizes the most recent GBD 2021 dataset to provide a comprehensive assessment of the global stroke burden attributable to HAP. Specifically, we analyze the burden stratified by age, gender, Socio-demographic Index (SDI), country, and region in 2021. We also examine spatiotemporal trends from 1990 to 2021 to elucidate changes in disease burden and its determinants. Unlike prior studies, we extend the analysis by incorporating detailed trends in years lived with disability (YLDs), which have shown a distinct increase, and by projecting the global stroke burden related to HAP through 2040 using the Nordpred model, offering a longer predictive horizon than previously available. This study not only updates and extends prior findings but also provides actionable knowledge for addressing a critical and preventable risk factor for stroke on a global scale.

## 2. Methods

### 2.1. Data source

**2.1.1. Data collection.** Annual data were obtained on age-standardized rates (ASRs), deaths, disability-adjusted life years (DALYs), years lived with disability (YLDs), and years of life lost (YLLs) due to stroke attributable to HAP from the Global Health Data Exchange GBD Results Tool (http://ghdx.healthdata.org/gbd-results-tool). The data were categorized by sex and age, covering global, regional, and national levels from 1990 to 2021. This tool, established by GBD collaborators, aims to quantify the relative severity of health loss (age- and sex-specific) caused by 369 diseases and injuries and 87 risk factors across 204 countries and territories for comprehensive epidemiological data assessment.

**2.1.2. Socio-demographic Index (SDI) and regional classification.** The SDI, calculated by integrating lagged distributed income per capita, average educational attainment, and total fertility rate, classified the 204 countries and territories into high (>0.81), high-middle (0.70–0.81), middle (0.61–0.69), low-middle (0.46–0.60), and low (<0.46) SDI regions Table 1. Additionally, these countries and territories were grouped into 21 geographical regions according to geographical location.

**2.1.3. Age stratification.** In this study, it is assumed that the stroke burden associated with HAP in the population under 25 years old is negligible, consistent with previous epidemiological studies that show stroke incidence increases substantially with age and is relatively rare in younger populations [16–18]. Therefore, for stratified analysis, the population aged 25 and above was divided into 15 age groups: 25–29, 30–34, 35–39, 40–44, 45–49, 50–54, 55–59, 60–64, 65–69, 70–74, 75–79, 80–84, 85–89, 90–94, and 95+years old.

**2.1.4. Ethics statement.** This study used publicly available, de-identified data from the Global Burden of Disease (GBD) 2021 database. As no human participants were directly involved and no identifiable personal information was used, ethical approval and informed consent were not required.

### 2.2. Measurements

Global stroke deaths, DALYs, YLDs, and YLLs attributable to HAP were reported. DALYs quantify overall disease burden, expressed as the number of years lost due to ill health, disability, or early death. It comprises two components: YLDs, which are years lived with any short-term or long-term health loss, weighted by severity using disability weights; and YLLs, calculated by multiplying observed deaths for a specific age in a particular year by the reference life expectancy at that age, estimated using life table methods [19].

The SDI is a composite indicator developed by the GBD study to measure the level of socioeconomic development of a region or country. It combines information from three main aspects: educational attainment (measured by mean years of education for the population aged 25 and older), fertility (represented by the total fertility rate among women aged 15–49 years), and income level (reflected by the logarithm of per capita gross national income). SDI values range from 0 to 1, with 0 denoting the lowest level of social development and 1 denoting the highest [20].

**Table 1. Global Trends in HAP-Induced Stroke Outcomes: EAPC Analysis Across 21 Regions for Deaths, DALYs, YLDs, and YLLs (1990-2021).**

| Region | EAPC in Deaths (95% CI) | EAPC in DALYs (95% CI) | EAPC in YLDs (95% CI) | EAPC in YLLs (95% CI) |
|---|---|---|---|---|
| East Asia | 1.67 (1.17, 2.17) | 1.58 (1.13, 2.03) | 3.95 (3.57, 4.33) | 1.42 (0.95, 1.88) |
| Southeast Asia | 0.64 (0.49, 0.80) | 0.46 (0.31, 0.61) | 0.72 (0.57, 0.86) | 0.44 (0.29, 0.60) |
| Oceania | −0.05 (−0.23, 0.13) | −0.08 (−0.26, 0.10) | 0.45 (0.30, 0.60) | −0.11 (−0.30, 0.07) |
| Central Asia | 0.52 (0.14, 0.90) | 0.28 (−0.07, 0.63) | 1.34 (0.86, 1.82) | 0.19 (−0.15, 0.54) |
| Central Europe | −3.77 (−4.08, −3.46) | −3.82 (−4.13, −3.51) | −1.79 (−2.03, −1.55) | −3.98 (−4.30, −3.66) |
| Eastern Europe | −5.43 (−5.88, −4.97) | −5.20 (−5.66, −4.73) | −3.24 (−3.37, −3.11) | −5.33 (−5.82, −4.84) |
| High-income Asia Pacific | −4.51 (−4.88, −4.14) | −3.95 (−4.32, −3.59) | −1.21 (−1.56, −0.87) | −4.57 (−4.92, −4.22) |
| Australasia | −2.99 (−3.51, −2.47) | −2.72 (−3.28, −2.16) | −0.37 (−0.87, 0.13) | −3.21 (−3.77, −2.65) |
| Western Europe | −6.28 (−6.47, −6.10) | −6.04 (−6.22, −5.86) | −3.53 (−3.78, −3.27) | −6.46 (−6.65, −6.27) |
| Southern Latin America | −3.61 (−3.83, −3.39) | −3.71 (−3.94, −3.48) | −1.83 (−2.04, −1.61) | −3.94 (−4.17, −3.71) |
| High-income North America | −5.42 (−5.88, −4.96) | −5.19 (−5.63, −4.74) | −4.14 (−4.68, −3.60) | −5.46 (−5.89, −5.03) |
| Caribbean | −0.71 (−0.87, −0.55) | −0.62 (−0.81, −0.43) | 0.13 (−0.02, 0.28) | −0.66 (−0.85, −0.46) |
| Andean Latin America | −3.89 (−4.25, −3.53) | −3.94 (−4.28, −3.60) | −2.29 (−2.63, −1.94) | −4.06 (−4.40, −3.71) |
| Central Latin America | −4.01 (−4.23, −3.80) | −3.88 (−4.09, −3.67) | −2.84 (−3.07, −2.60) | −3.97 (−4.18, −3.75) |
| Tropical Latin America | −3.62 (−3.84, −3.39) | −3.80 (−4.02, −3.58) | −1.96 (−2.20, −1.72) | −3.89 (−4.11, −3.67) |
| North Africa and Middle East | −0.81 (−0.93, −0.70) | −0.88 (−1.01, −0.75) | 0.47 (0.31, 0.62) | −0.96 (−1.09, −0.83) |
| South Asia | 1.72 (1.33, 2.11) | 1.63 (1.26, 2.00) | 2.45 (2.10, 2.81) | 1.58 (1.21, 1.95) |
| Central Sub-Saharan Africa | 0.44 (0.20, 0.68) | 0.41 (0.17, 0.64) | 1.04 (0.76, 1.32) | 0.37 (0.14, 0.60) |
| Eastern Sub-Saharan Africa | −0.06 (−0.22, 0.09) | −0.22 (−0.38, −0.06) | 1.01 (0.83, 1.18) | −0.29 (−0.45, −0.13) |
| Southern Sub-Saharan Africa | 0.90 (0.47, 1.33) | 0.53 (0.13, 0.93) | −0.25 (−0.35, −0.15) | 0.60 (0.17, 1.04) |
| Western Sub-Saharan Africa | 0.18 (−0.21, 0.57) | 0.03 (−0.34, 0.41) | 0.99 (0.57, 1.41) | −0.03 (−0.40, 0.34) |

EAPC: Estimated Annual Percentage Change; CI: Confidence Interval; DALYs: Disability-Adjusted Life Years; YLDs: Years Lived with Disability; YLLs: Years of Life Lost.

### 2.3 Statistical analysis

**2.3.1. Time trend analysis.** The time trends of stroke burden caused by HAP from 1990 to 2019 were explored globally and reported by different subtypes, including age, sex, SDI, region, and country. The Estimated Annual Percentage Change (EAPC) value was calculated to quantify trends over time. The EAPC value was calculated using a log-linear regression model. A positive EAPC value indicates an increasing trend, while a negative value indicates a decreasing trend. The magnitude of the EAPC reflects the rate of change, with larger absolute values indicating faster change.

**2.3.2. Cluster analysis.** Based on the EAPC values, hierarchical clustering analysis was conducted to assess the change patterns of stroke burden attributed to HAP across different GBD regions and to identify areas with similar changes in disease burden.

**2.3.3. Correlation analysis.** To explore the factors influencing changes in disease burden, we calculated the Pearson correlation between ASRs related to HAP and the SDI globally and across 21 regions from 1990 to 2021.

**2.3.4. Future projections.** The Nordpred prediction model was used to forecast future disease burden from 2022 to 2040. This model is an improved version of the Age-Period-Cohort (APC) regression framework, specifically designed for predicting trends in health outcomes. All statistical analyses and visualizations were performed using R statistical software (version 4.3.1). $P$ value $< 0.05$ was considered statistically significant.

## 3. Results

### 3.1 The burden of stroke attributable to indoor air pollution and its temporal trends

**3.1.1. Global burden in 2021.** In 2021, the number of stroke deaths related to HAP was 1,230,852 [95% uncertainty interval (UI): 834,767-1,575,049], with 26,779,430 DALYs (95% UI: 18,076,188–34,223,093), 2,425,770 YLDs (95% UI: 536,922–3,438,131), and 24,353,661 YLLs (95% UI: 16,319,005–31,243,808) (S1 Table in S1 File).

**3.1.2. Temporal trends (1990–2021).** From 1990 to 2021, the global age-standardized mortality rate attributed to HAP decreased by 0.37 per year, from 17.42 per 100,000 to 14.78 per 100,000. In terms of the DALYs rate, the global age-standardized rate decreased from 349.70 per 100,000 in 1990 to 312.85 per 100,000 in 2021, with an EAPC of −0.20 [95% Confidence Interval (CI): −0.34 to −0.07]. The age-standardized YLLs rate decreased from 327.81 per 100,000 in 1990 to 284.63 per 100,000 in 2021, with an EAPC of −0.30 (95% CI: −0.44 to −0.17). However, the age-standardized YLDs rate showed an increasing trend, rising from 21.89 per 100,000 in 1990 to 28.22 per 100,000 in 2021, with an EAPC of 1.11 (95% CI: 0.94 to 1.27) (S1 Table in S1 File, Fig 1).

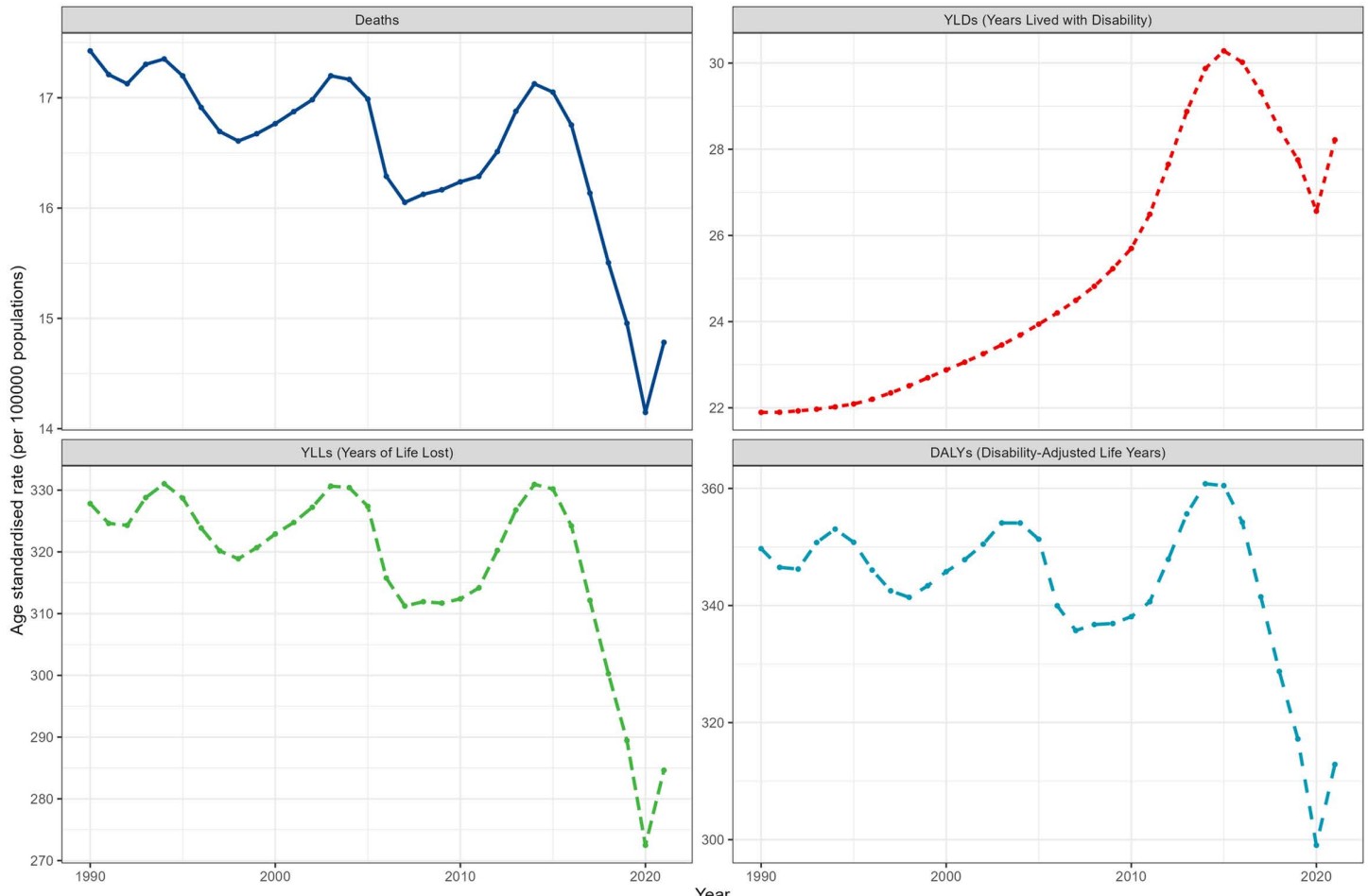

**Fig 1. Temporal trends of age-standardized stroke burden indicators attributable to household air pollution (1990-2021).** All metrics are presented as age-standardized rates per 100,000 population.

### 3.2 Variation in HAP-induced stroke burden in two sexes

In 2021, the number of deaths, DALYs, YLDs, and YLLs for males were 1.25 times, 1.38 times, 1.02 times, and 1.42 times that of females, respectively. The corresponding ASRs were 1.59 times, 1.61 times, 1.15 times, and 1.66 times, respectively (S2 Table in S1 File).

The ASRs for males all showed an increasing trend, with the age-standardized YLDs rate increasing the most significantly (1.17, 95% CI: 1.00 to 1.13). In contrast, the ASRs for females showed a significant decrease, except for a substantial increase in the age-standardized YLDs rate (1.02, 95% CI: 0.85 to 1.19) (S2 Table in S1 File, Fig 2).

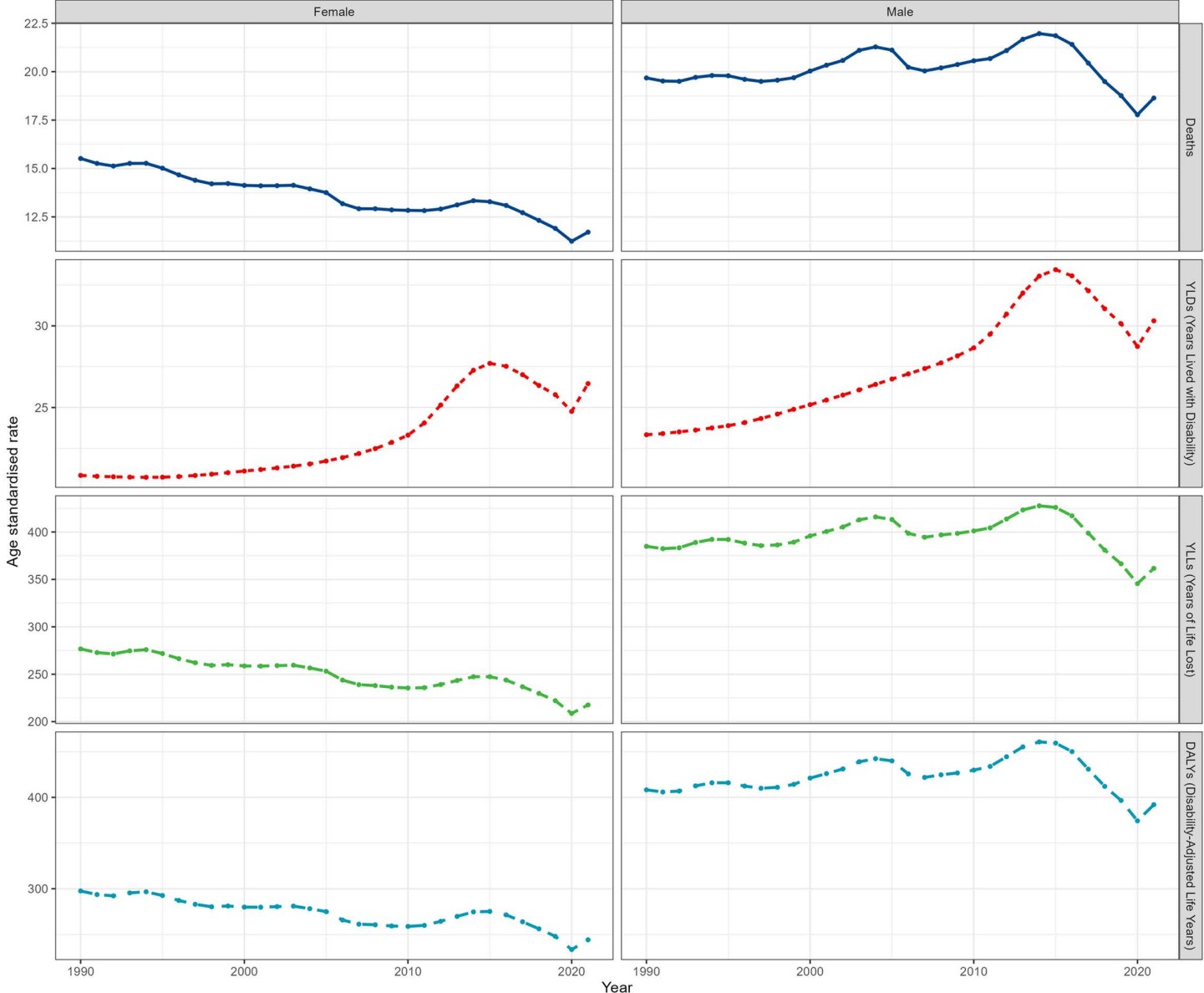

**Fig 2. Gender disparities in stroke burden attributable to household air pollution from 1990 to 2021: dynamic changes in deaths, YLDs, YLLs, and DALYs.**

### 3.3 Variation in HAP-induced stroke burden in five-year age groups

Fig 3 shows the number of deaths, DALYs, YLDs, and YLLs for each age group from 1990 to 2021. The age-standardized mortality rate first increases with age, peaking in the 80–84 age group, and then declines. The age-standardized rates for DALYs, YLDs, and YLLs follow the same age pattern as the age-standardized mortality rate (Fig 3).

### 3.4 Variation in HAP-induced stroke burden by SDI

At the SDI regional level, the middle SDI region had the highest age-standardized mortality rate, DALYs rate, YLDs rate, and YLLs rate (Table S4 in S1 File). Furthermore, the age-standardized rates (ASRs) of deaths, DALYs, and YLLs increased in the middle and low-middle SDI regions, while these metrics decreased in the high, high-middle, and low SDI regions. In contrast, the ASR of YLDs showed a distinct pattern, increasing across all SDI regions except the high SDI region, where a decrease was observed (S3 Table in S1 File, Fig 4).

### 3.5 Variation in HAP-induced stroke burden at regional level

**3.5.1. Regional burden in 2021.** Among the 21 GBD regions, East Asia ranked first in stroke deaths, DALYs, YLDs, and YLLs related to HAP, Australasia ranked last in DALYs and YLLs, while Oceania ranked last in deaths and YLDs

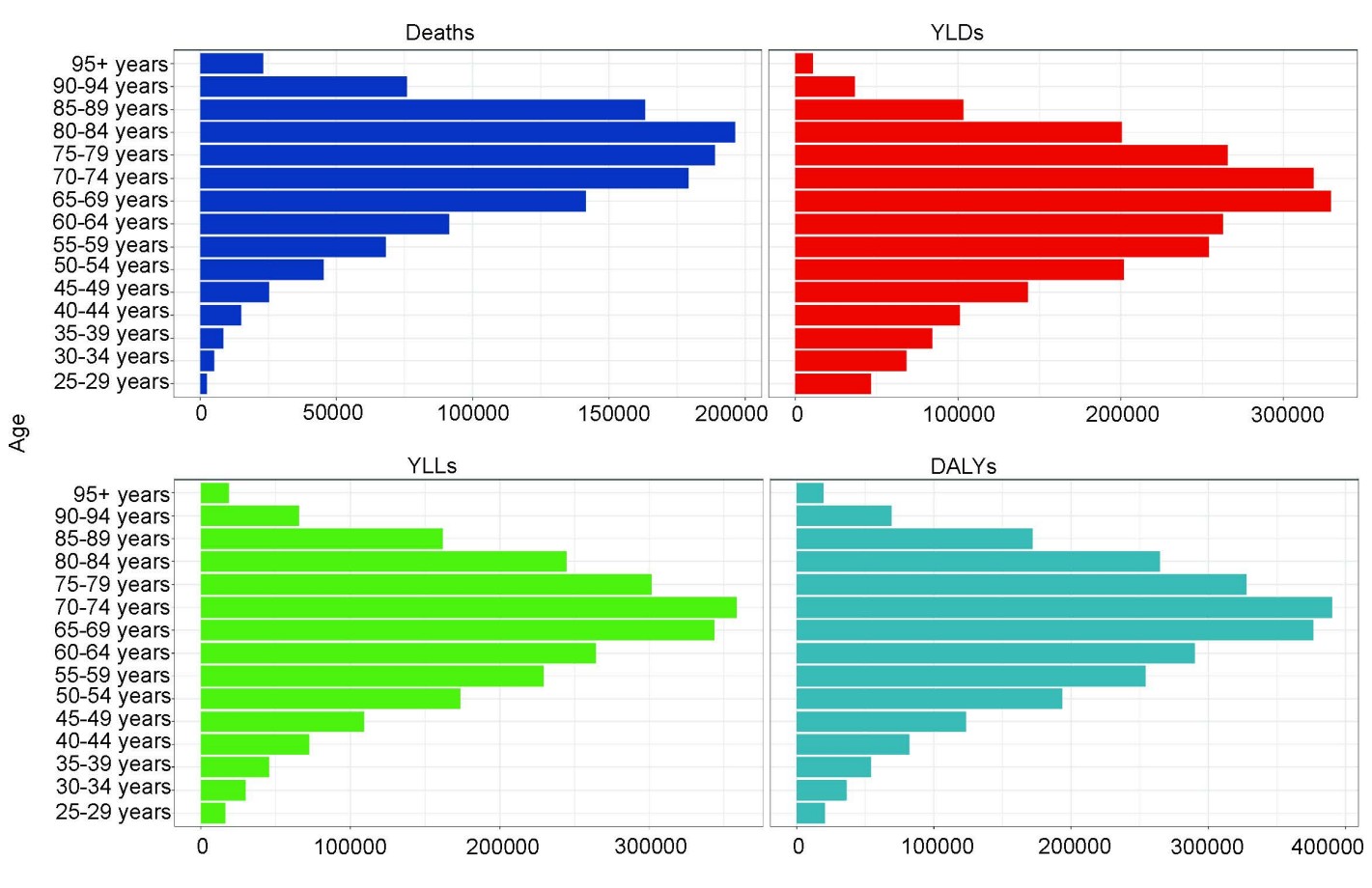

**Fig 3. Numbers and age-standardized rates of household air pollution-induced stroke-related Deaths, YLDs, YLLs, and DALYs for different age groups in 2021.**

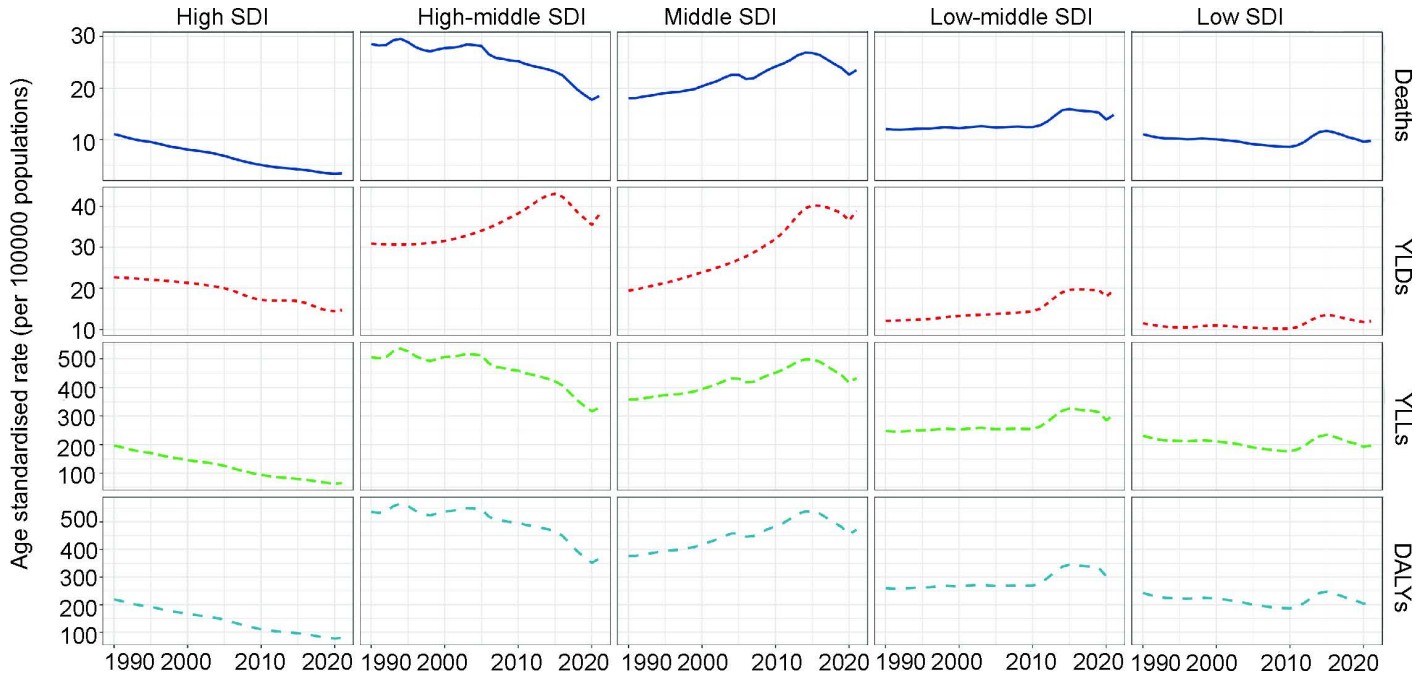

**Fig 4. Trends of Deaths, YLDs, YLLs, and DALYs of stroke in regions with different development levels from 1990 to 2021.** Data are stratified by five SDI levels: high, high-middle, middle, low-middle, and low.

(Table S4 in S1 File). Among the corresponding ASRs, the highest-ranking GBD region is East Asia and the lowest is high-income North America. (Table S5 in S1 File). The disease burden related to HAP varies across GBD regions.

**3.5.2. Regional trends and clustering.** This study conducted a hierarchical clustering analysis to identify regions with similar disease burden changes. The regions with the greatest decline in mortality rates are Western Europe, Eastern Europe, and High-income North America. The most significant increases in mortality rates were observed in South Asia and East Asia (Fig 5A, Table S5 in S1 File). The DALYs and YLLs rates followed the same pattern as the mortality rates (Fig 5B–5C, Table S5 in S1 File). Unlike other metrics, YLDs showed a different pattern. The fastest growth was seen in East Asia, followed by South Asia. High-income North America and Western Europe experienced the largest declines. Although Western Europe still showed a downward trend, the decline was not as pronounced as for other metrics (Fig 5D, S5 Table in S1 File).

**3.5.3. Correlation with SDI.** Fig 6A shows no correlation between age-standardized mortality rate and SDI ($\rho = -0.02$, $P = 0.452$). DALYs had a weak positive correlation with SDI, but it was not statistically significant ($\rho = 0.06$, $P = 0.061$) (Fig 6B). YLLs had a weak positive correlation with SDI ($\rho = 0.07$, $P = 0.017$) (Fig 6C). YLDs had a positive correlation with SDI ($\rho = 0.27$, $P < 0.001$) (Fig 6D). With the increase in SDI, the ASRs of stroke caused by HAP exhibit an inverted U-shaped pattern, initially rising and then falling. Some regions deviate from the overall trend line. Specifically, Sub-Saharan Africa is characterized by a low development index and a high burden. Most countries in South Asia and Southeast Asia are at a medium level of development but have a relatively high burden. In East Asia, particularly China, there is a notable trajectory of reducing burden levels significantly, along with rapid growth in SDI. High-income regions (including Western Europe, North America, and the High-income Asia-Pacific) are located in the lower right of the chart, indicating the highest SDI levels and the lowest stroke burden related to HAP.

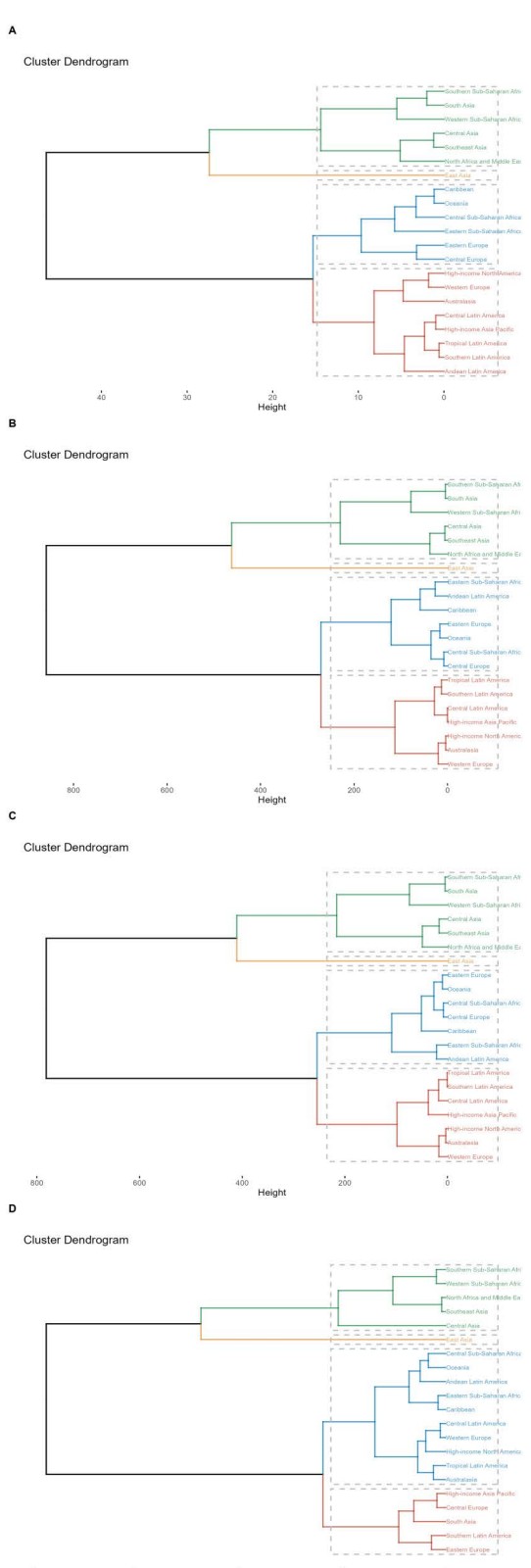

**Fig 5. Results of cluster analysis based on EAPC values of HAP-related age-standardized mortality rates, DALYs, YLDs, and YLLs from 1990 to 2021.** (A) Deaths; (B) DALYs; (C) YLLs; (D) YLDs.

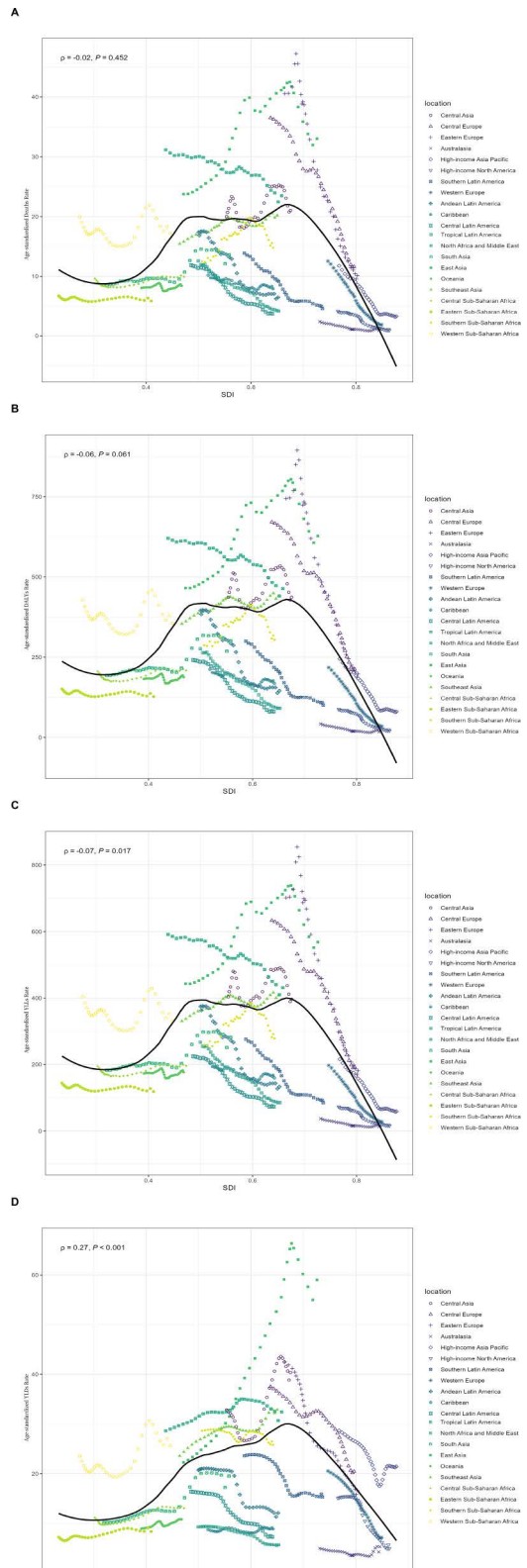

**Fig 6. Regional variations in stroke burden attributable to household air pollution across different SDI levels from 1990 to 2021.** (A) Association between age-standardized stroke mortality rate and SDI; (B) Association between age-standardized stroke DALYs and SDI; (C) Association between

age-standardized stroke YLLs and SDI; (D) Association between age-standardized stroke YLDs and SDI. Each colored line represents a different global region, while the black line indicates the overall trend. The provided ρ coefficients and p-values are derived from Spearman correlation analysis.

### 3.6 Variation in HAP-induced stroke burden at national and territorial level

**3.6.1. National burden in 2021.** The stroke burden due to HAP varies significantly across the world. In 2021, Egypt had the highest age-standardized mortality rate (66.26, 95% UI: 46.68–86.59), age-standardized DALYs rate (1222.61, 95% UI: 864.22–1588.03), and YLLs rate (1165.88, 95% UI: 812.57–1528.54), followed by North Macedonia and Iraq (Fig 7A–7C). The highest age-standardized YLD rate was observed in China (60.32, 95% UI: 36.40–85.64), followed by Egypt and Uzbekistan (Fig 7D).

**3.6.2. National trends (1990–2019).** Among all countries and regions, Vietnam had the greatest increase in mortality and YLL burden for ASRs from 1990 to 2019 (Mortality: 4.95, 95% CI: 4.60–5.29; YLLs: 4.79, 95% CI: 4.43–5.15), followed by Bhutan and Cape Verde. Estonia showed the greatest decline (Mortality: −10.72, 95% CI: −11.43 to −10.00; YLLs: −10.86, 95% CI: −11.58 to −10.14), followed by Norway and Portugal. The greatest increase in DALY burden was observed in Vietnam (4.81, 95% CI: 4.46 to 5.17), followed by Mongolia and Cape Verde. Estonia showed the greatest decline (−10.33, 95% CI: −10.98 to −9.66), followed by Norway and Portugal. The largest increase in YLD burden was in Mongolia (5.52, 95% CI: 4.90 to 6.14), followed by Vietnam and Equatorial Guinea. Estonia showed the greatest decline (−5.78, 95% CI: −6.11 to −5.46), followed by Norway and Finland (S6 Table in S1 File).

### 3.7 Future burden of HAP-induced stroke

Fig 8 illustrates the observed and projected trends of stroke burden attributable to hHAP from 1990 to 2040, stratified by sex (males, females, and both sexes combined). Each metric—deaths, YLDs, YLLs, and DALYs—is presented in absolute case numbers per 100,000 people and ASRs. The projections indicate a continuous increase in the number of deaths, YLDs, YLLs, and DALYs from 2022 to 2040, but the corresponding ASRs are declining. For both males and females, the absolute number of deaths, YLDs, YLLs, and DALYs is projected to increase from 2021 to 2040, consistent with the overall trend. However, males consistently exhibit a higher burden across all metrics than females, a pattern that persists into future projections. For instance, the absolute number of deaths among males is projected to rise more steeply than among females. Similarly, DALYs and YLLs for males are expected to increase at a faster rate than for females, while YLDs show a more pronounced rise for both sexes, though still higher in males. Despite these increases in absolute numbers, the ASRs for both sexes are projected to decline by 2040 (S7 Table in S1 File, Fig 8).

## 4. Discussion

To our knowledge, this is the first study to comprehensively assess the global stroke burden related to HAP using GBD 2021 data and to predict future trends. In 2021, HAP imposed a significant disease burden globally, with notable differences across genders, age groups, SDI regions, GBD regions, and countries. From 1990 to 2021, although the global ASRs showed a declining trend, the absolute numbers of deaths, YLDs, YLLs, and DALYs remained significant and showed an increasing trend. Furthermore, our projections indicate that over the next 19 years, the absolute numbers of deaths, YLDs, YLLs, and DALYs will continue to rise.

Some previous studies have estimated the stroke burden related to HAP. However, these studies were mostly limited to individual regions or countries [14]. Only a few studies have been conducted on a global scale [16]. For example, Shi et al. found that HAP generated by the use of solid fuels is associated with a high risk of chronic comorbidities (such as stroke, cardiovascular diseases, and hypertension) among Chinese adults [21]. This study used three waves of data, including 19,295 participants aged ≥45 years from the China Health and Retirement Longitudinal Study. Since this cohort

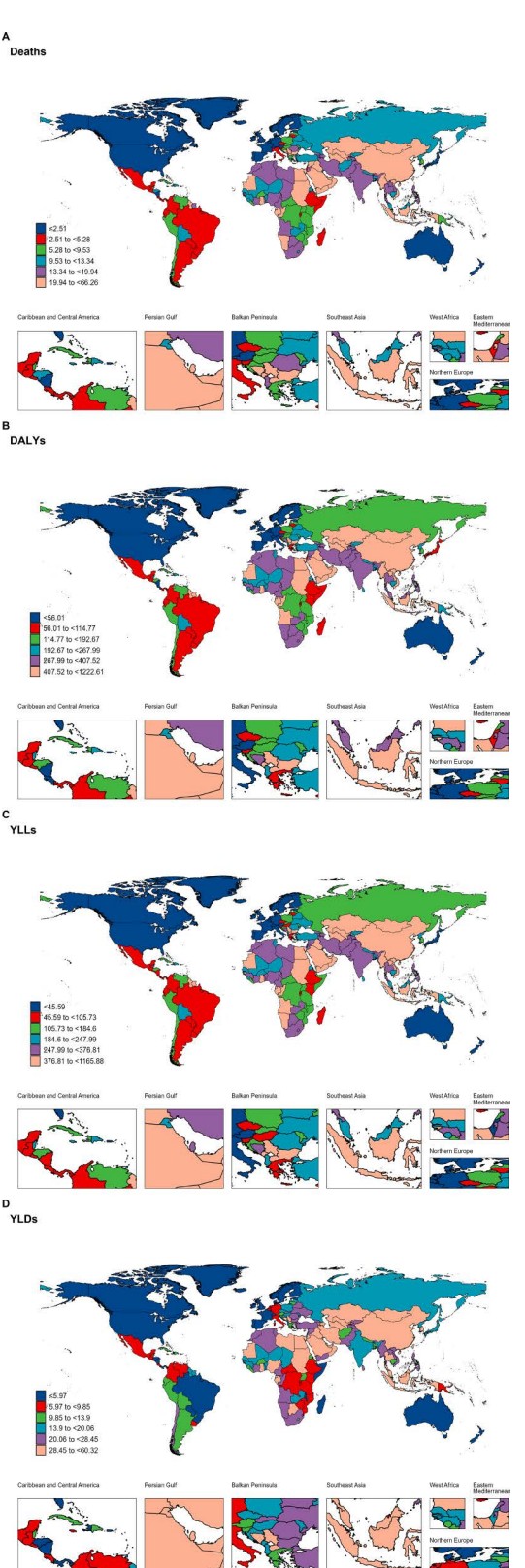

**Fig 7. Global burden of disease of stroke attributed to household air pollution in 204 countries and territories.** (A) Deaths; (B) DALYs; (C) YLLs; (D) YLDs.

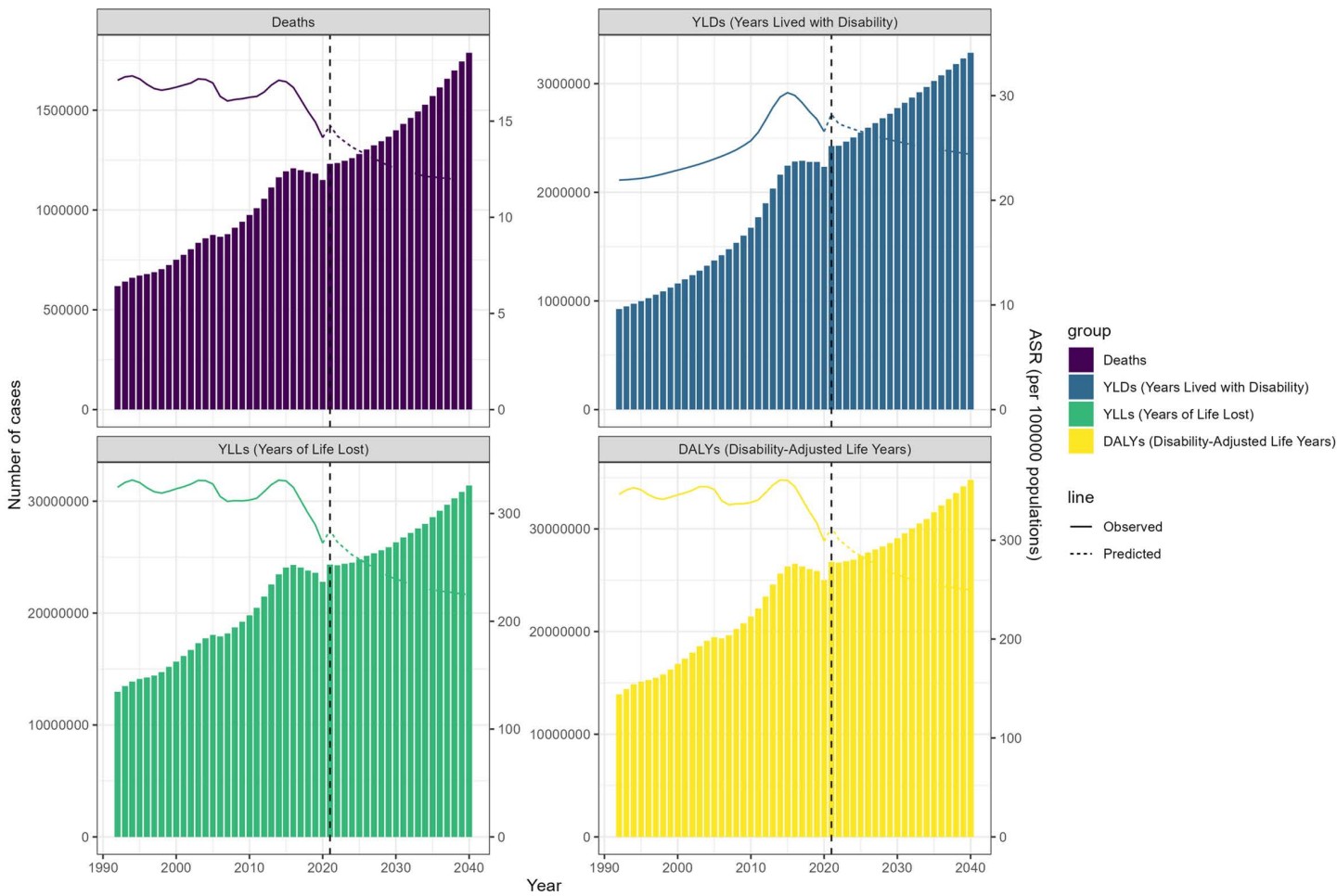

**Fig 8. Future Forecasts of GBD in Household Air Pollution-Induced Stroke.**

was conducted only in localized areas, there may be some limitations, leading to insufficient accuracy in methodological extrapolation. In contrast, the GBD 2021 study data are derived from multiple sources, such as household surveys, vital statistics, and other datasets. This study was conducted in numerous countries. Therefore, the GBD study provides more accurate estimates of the disease burden. Lu et al. [16] used GBD 2019 study data to explore the spatiotemporal trends of mortality and stroke burden related to HAP from 1990 to 2019, reporting 0.6 million stroke deaths and 14.7 million DALYs in 2019. While their work laid a critical foundation, it is now outdated and did not capture the increasing trend in YLDs, nor did it provide projections beyond 2019. Our study substantially extends this prior work by leveraging the updated GBD 2021 dataset, which includes two additional years of data and refined estimates, revealing a higher burden (1.23 million deaths and 26.78 million DALYs in 2021). We also highlight a significant increase in YLDs (EAPC = 1.11), contrasting with declining mortality and YLL trends, shedding new light on the growing disability burden of HAP-related stroke. Furthermore, using the Nordpred model, we project trends to 2040, offering a forward-looking perspective essential for long-term policy planning.

Between 1990 and 2021, the age-standardized burden of stroke related to HAP exhibited an overall declining trend, likely attributable to global efforts to promote improved stoves and clean energy adoption [22,23]. This reduction reflects

progress in lowering HAP exposure relative to population and age structure. However, absolute numbers of deaths, YLDs, YLLs, and DALYs rose, driven by population growth and aging [24,25]. In low- and middle-income regions, where HAP remains common, a growing population increases the number at risk, elevating the total burden despite falling per-capita rates. Concurrently, aging populations—particularly in areas with rising life expectancy—face higher stroke rates due to prolonged HAP exposure and age-related cardiovascular vulnerability [24]. Urbanization and industrial growth further counteract gains in age-standardized metrics [26,27]. Thus, while interventions have reduced the relative burden, the absolute number of cases continues to climb, highlighting the need for sustained, targeted action against this preventable risk factor.

We found that the stroke burden attributable to HAP is higher in males than in females. This gender disparity can be attributed to differences in exposure levels and lifestyle behaviors [15,28]. A significant factor contributing to the elevated stroke burden in men is the higher prevalence of risk factors linked to HAP exposure, such as smoking [29,30]. In addition, men often have higher rates of other cardiovascular risk factors, such as hypertension and elevated body mass index (BMI), which can exacerbate the effects of HAP exposure [31,32]. Further, occupational exposure may partially explain this disparity [33,34]. Men are more likely to engage in occupations with high ambient air pollution exposure (e.g., mining, construction, or industrial work), which could compound the effects of household air pollution [35,36]. Although the GBD data do not explicitly account for occupational exposures, this synergistic effect may amplify the overall risk of stroke in males. Additionally, cultural norms in some regions may lead to longer indoor exposure times for women during cooking; however, men's higher susceptibility to smoking and comorbidities might offset this behavioral difference [37]. Future studies should incorporate occupational exposure data to disentangle these complex interactions.

The burden of stroke and mortality attributable to HAP exhibits a trend of initially increasing and then decreasing with age. As individuals age, the cumulative effects of HAP exposure and other cardiovascular risk factors may result in an increased stroke burden [38]. Evidence indicates that older adults are particularly susceptible to the effects of HAP due to their heightened vulnerability to respiratory and cardiovascular diseases resulting from exposure to PM [39]. However, over time, improvements in public health interventions—such as raising awareness about the dangers of HAP and promoting cleaner cooking technologies—may help mitigate the stroke burden associated with HAP in older populations [14,29]. Additionally, survivorship bias may contribute to the lower observed stroke burden in older age groups [40].

Stroke burden related to HAP varies significantly across regions and countries, driven by differences in healthcare systems, living environments, and socioeconomic conditions [41,42]. South Asia and East Asia have seen the fastest increases in HAP-related stroke burden, while Western Europe, Eastern Europe, and high-income North America exhibit the greatest declines. Notably, our study found that middle-SDI regions had the highest age-standardized rates of stroke mortality, DALYs, YLDs, and YLLs in 2021, contrasting with prior research emphasizing low-SDI regions as most affected by air pollution. This discrepancy may arise from underreporting and limited healthcare access in low-SDI regions, masking their true burden in GBD estimates, whereas middle-SDI regions, like parts of East Asia and South Asia, face a dual burden: ongoing use of solid fuels and rising ambient pollution from rapid urbanization [37,43]. Lacking the infrastructure for full clean-energy adoption—unlike high-SDI regions—these areas experience heightened stroke risk.

The divergent trends in specific countries further illustrate the role of policy and environmental factors. Vietnam and Mongolia experienced the most dramatic increases in HAP-related stroke burden, while Estonia, Norway, and Finland achieved the most substantial reductions. These disparities arise from multiple factors. Primarily, energy transition policies are pivotal; nations with declining stroke burdens often adopt robust clean energy initiatives and subsidies, shifting from solid fuels to alternatives like electricity, natural gas, or renewables [44]. Additionally, economic development shapes HAP exposure, with Estonia and Nordic countries achieving equitable growth that enhances access to clean cooking technologies across income levels. Moreover, rigorous indoor air quality regulations and enforcement distinguish countries with

reduced HAP-related stroke burdens. Lastly, geographic and climatic factors influence heating and ventilation needs; colder regions like Estonia and Norway mitigate HAP through modernized heating systems despite high demand [45].

One notable observation from our analysis is the divergence of YLD trends from other indices, particularly in middle and high-middle SDI countries, as illustrated in Fig 4. While age-standardized mortality, DALY, and YLL rates have generally declined across SDI levels from 1990 to 2021, the YLD rate in high-middle SDI regions remained stable or slightly increased. This discrepancy may reflect improvements in healthcare systems and stroke management in these transitional economies, where increased survival rates following stroke lead to a higher prevalence of individuals living with long-term disabilities [46]. Unlike high SDI regions, where both prevention and rehabilitation are more advanced, high-middle SDI countries may be in a phase of rapid development that reduces mortality but not disability. Additionally, residual HAP exposure, combined with emerging risk factors such as aging populations or dietary shifts, could sustain stroke-related disability burdens in these regions.

The projections indicate that from 2021 to 2040, the numbers of deaths, YLDs, YLLs, and DALYs may increase to varying extents. However, the corresponding ASRs are expected to show a declining trend. This may be attributed to the combined effects of future demographic changes, increased high-risk behaviors, and socioeconomic development. The gender-specific trends further highlight that while both sexes will experience an increase in absolute burden, males will bear a greater share, underscoring the interplay of biological, behavioral, and environmental factors.

However, some limitations should be considered when interpreting the results. First, although the database is comprehensive, the varying quality and availability of data between countries and regions may introduce uncertainty when using models to estimate disease burden [47]. Second, the stroke burden attributable to HAP may be underestimated due to the assumption that there is no interaction between HAP and ambient air pollution. In reality, HAP and ambient air pollution may exhibit synergistic effects, with their combined impact potentially exceeding the simple sum of their independent effects [48]. Third, the observed trends of decreasing age-standardized mortality, DALYs, and YLLs, alongside an increasing YLDs trend, may be significantly influenced by evolving treatment modalities for stroke, representing an important confounding factor. Advances in medical interventions, such as improved acute stroke care (e.g., thrombolysis and endovascular therapy), better secondary prevention strategies (e.g., antihypertensive and statin therapies), and enhanced rehabilitation programs, have likely reduced mortality and severe disability while increasing the number of individuals living with milder, long-term disabilities. These developments could disproportionately affect the burden metrics, particularly in high-SDI regions where access to such treatments is more widespread. Despite these limitations, the GBD's data inclusion and methodological standards are consistent across all countries, thereby reducing biases when comparing metrics across nations. Future research should prioritize improving data collection methods and refining exposure assessment techniques to enhance understanding of the stroke burden attributable to HAP and inform more effective prevention and intervention strategies.

## 5. Conclusions

HAP remains a major global stroke risk factor, disproportionately affecting middle-SDI regions, men, and older adults. While age-standardized rates are projected to decline over the next 19 years, the absolute burden will rise, with males consistently facing a higher toll than females due to exposure and risk factor disparities. These trends underscore HAP as an urgent public health priority, necessitating gender-sensitive strategies and stricter interventions to reduce exposure and protect vulnerable populations.

## Supporting information

**S1 File. Supplementary Material.** Global Stroke Burden Attributable to Household Air Pollution: Insights from GBD 2021 and Projections to 2040.
(DOCX)

**S2 File. Graphical Abstract. Global Burden of Disease (GBD) Analysis for Household Air Pollution (HAP)-Induced Stroke.** This graphical abstract illustrates the relationship between HAP and stroke, based on GBD data. The left panel depicts key pollutants from HAP, such as particulate matter (PM), carbon monoxide (CO), nitrogen oxides ($NO_x$), and sulfur dioxide ($SO_2$), which contribute to stroke. The middle section shows the progression to stroke, while the lower section highlights the influence of factors such as age, sex, and socio-demographic index (SDI). The right panel presents global spatial patterns of stroke-related mortality, Disability-Adjusted Life Years (DALYs), Years Lived with Disability (YLDs), and Years of Life Lost (YLLs) in 2021, temporal trends from 1990 to 2021, and projections for the next 19 years.
(TIF)

## Acknowledgments

The authors sincerely appreciate all the participants of the GBD 2021 for their contribution.

## Author contributions

**Data curation:** Manqing Li.

**Formal analysis:** Zhuoying Zeng.

**Funding acquisition:** Jie Jiang, Zhuoying Zeng.

**Methodology:** Manqing Li, Jie Jiang.

**Project administration:** Jie Jiang, Zhuoying Zeng.

**Resources:** Jie Jiang, Zhuoying Zeng.

**Software:** Manqing Li.

**Visualization:** Manqing Li.

**Writing – original draft:** Manqing Li.

**Writing – review & editing:** Shicheng Liao, Chao Wang, Haojia Ma, Xiumei Xing, Jianhui Yuan, Jie Jiang, Zhuoying Zeng.

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
