## [Decision Letter · Decision Letter 0]

PONE-D-25-02181Global Stroke Burden Attributable to Household Air Pollution: Insights from GBD 2021 and Projections to 2040PLOS ONE

Dear Dr. Zeng,

Thank you for submitting your manuscript to PLOS ONE. After careful consideration, we feel that it has merit but does not fully meet PLOS ONE’s publication criteria as it currently stands. Therefore, we invite you to submit a revised version of the manuscript that addresses the points raised during the review process.

We look forward to receiving your revised manuscript.

Kind regards,

Amir Hossein Behnoush

Academic Editor

PLOS ONE

Journal Requirements:

3. Please note that funding information should not appear in the Acknowledgments section or other areas of your manuscript. We will only publish funding information present in the Funding Statement section of the online submission form. Please remove any funding-related text from the manuscript. 

4. We note that Figure 7 in your submission contain map images which may be copyrighted. All PLOS content is published under the Creative Commons Attribution License (CC BY 4.0), which means that the manuscript, images, and Supporting Information files will be freely available online, and any third party is permitted to access, download, copy, distribute, and use these materials in any way, even commercially, with proper attribution. For these reasons, we cannot publish previously copyrighted maps or satellite images created using proprietary data, such as Google software (Google Maps, Street View, and Earth). For more information, see our copyright guidelines: http://journals.plos.org/plosone/s/licenses-and-copyright.

1) You may seek permission from the original copyright holder of Figure 7 to publish the content specifically under the CC BY 4.0 license.  

2) If you are unable to obtain permission from the original copyright holder to publish these figures under the CC BY 4.0 license or if the copyright holder’s requirements are incompatible with the CC BY 4.0 license, please either i) remove the figure or ii) supply a replacement figure that complies with the CC BY 4.0 license. Please check copyright information on all replacement figures and update the figure caption with source information. If applicable, please specify in the figure caption text when a figure is similar but not identical to the original image and is therefore for illustrative purposes only.

Reviewers' comments:

Reviewer's Responses to Questions

**Comments to the Author**

1. Is the manuscript technically sound, and do the data support the conclusions?

Reviewer #1: Yes

Reviewer #2: Yes

Reviewer #3: Yes

2. Has the statistical analysis been performed appropriately and rigorously? 

Reviewer #1: Yes

Reviewer #2: Yes

Reviewer #3: Yes

3. Have the authors made all data underlying the findings in their manuscript fully available?

Reviewer #1: Yes

Reviewer #2: Yes

Reviewer #3: Yes

4. Is the manuscript presented in an intelligible fashion and written in standard English?

Reviewer #1: Yes

Reviewer #2: Yes

Reviewer #3: Yes

5. Review Comments to the Author

Reviewer #1: The authors present a valuable study on the epidemiology of stroke attributed to household air pollution. Their findings indicate decreasing mortality, DALY, and YLL trends but an increasing YLD trend after age adjustment. However, these results may be significantly influenced by evolving treatment modalities that reduce both stroke-related mortality and morbidity, representing an important confounding factor. This limitation should be explicitly discussed.

Additionally, in Figure 4, the YLD trend does not align with other indices specifically in high-middle SDI countries, which warrants further explanation.The study may also benefit from discussing potential future trends based on geographic distribution, such as SDI categories, to provide a more comprehensive perspective.

Finally, the authors should ensure that the image resolution is sufficient, as the current version is not fully understandable.

Reviewer #2: this manuscript is novel and useful for the current literature, since it is debating on an association that has been newly described in literature. The addition of projection for 2040 by using Nordprod prediction model further approves the point.

Reviewer #3: The study provides a valuable update on the global stroke burden related to household air pollution (HAP), using the latest GBD 2021 data. However, while the projections to 2040 add a predictive aspect, it would be beneficial to clarify how this study substantially extends prior GBD-based research.

The manuscript is well-structured but dense with statistical details. Some sections, such as the methods and results, could benefit from improved readability by incorporating more subheadings.

Attached a document, may find it.

6. PLOS authors have the option to publish the peer review history of their article (what does this mean? ). If published, this will include your full peer review and any attached files.

**Do you want your identity to be public for this peer review?** For information about this choice, including consent withdrawal, please see our Privacy Policy .

Reviewer #1: No

Reviewer #2: **Yes: ** Alireza Ramandi

Reviewer #3: No

---

## [Author Response · Author response to Decision Letter 1]

23 Apr 2025

Response to Reviewers

1.Academic editor:

Reviewer point #1: We note that Figure 7 in your submission contains map images which may be copyrighted. We require you to either (1) present written permission from the copyright holder to publish these figures specifically under the CC BY 4.0 license, or (2) remove the figures from your submission.

Author response #1: We thank the reviewer for raising this important concern regarding Figure 7 in our manuscript. We would like to clarify that Figure 7, which displays the global burden of stroke attributed to household air pollution across 204 countries and territories, was entirely generated by our research team using R statistical software (version 4.3.1). These maps were created using open-source R packages (e.g., ggplot2, sf, or maps) and publicly available GBD 2021 data from the Global Health Data Exchange GBD Results Tool (http://ghdx.healthdata.org/gbd-results-tool), as referenced in our methods section. No copyrighted map images were incorporated or adapted from external sources. Since these visualizations were produced as original work by the authors using open data sources and software tools, they are not subject to external copyright restrictions and can be published under the CC BY 4.0 license as part of our manuscript. We hope this clarification adequately addresses the reviewer's concern.

Reviewer point #2: The study provides a valuable update on the global stroke burden related to household air pollution (HAP), using the latest GBD 2021 data. However, while the projections to 2040 add a predictive aspect, it would be beneficial to clarify how this study substantially extends prior GBD-based research.

Author response #2: We sincerely thank the reviewer for recognizing the value of our study and for providing constructive feedback. We agree that clarifying how our study extends prior GBD-based research is essential to highlight its novelty and significance. To address this comment, we have revised the manuscript in the following ways:

1)Introduction (Revised Paragraph):

We have explicitly compared our work to prior GBD-based studies, notably Lu et al. (2021), which used GBD 2019 data to estimate 600,000 stroke deaths and 14.7 million DALYs in 2019. We now emphasize that our study leverages the updated GBD 2021 dataset, incorporates detailed gender and age stratifications, highlights the unique increasing trend in years lived with disability (YLDs), and extends projections to 2040 using the Nordpred model. These additions clarify how our study goes beyond previous analyses by providing a more comprehensive and forward-looking assessment (see revised Introduction, second paragraph).

2)Discussion (Revised Paragraph):

In the Discussion, we further elaborate on the extensions of our study compared to Lu et al. (2021). We note that while their work was foundational, it is now outdated and did not address the significant rise in YLDs (EAPC=1.11) or provide long-term projections. Our study reports a higher burden (1.23 million deaths and 26.78 million DALYs in 2021) based on GBD 2021 data, offers new insights into the disability burden through YLDs analysis, and extends the temporal scope to 2040. These advancements provide a more complete picture of the evolving HAP-related stroke burden and its implications for future public health strategies (see revised Discussion, second paragraph).

These revisions strengthen the manuscript by clearly delineating how our study builds upon and extends prior GBD-based research, offering both updated estimates and novel predictive insights. We are grateful for this suggestion and welcome any further feedback.

Reviewer point #3: Abstract: The conclusion states that “higher economic development and adoption of clean energy reduce HAP-related stroke burden.” However, since this study is observational and not an interventional trial, the phrasing should clarify that this can be a hypothesis rather than a direct causation.

Author response #3: We appreciate the reviewer's thoughtful comment regarding the conclusion statement in our abstract. The reviewer correctly points out that our study is observational and not an interventional trial, so we should be cautious about implying causation. We agree that our original statement, "Higher economic development and adoption of clean energy reduce HAP-related stroke burden," should be rephrased to reflect the associational nature of our findings better rather than suggesting direct causation.

We have revised the conclusion in the abstract to clarify that our observation suggests an association between economic development, clean energy adoption, and reduced HAP-related stroke burden, rather than making a direct causal claim. The revised text now appropriately reflects the observational nature of our study while still conveying the important relationships we observed in the data.

Reviewer point #4: Abstract: The results mention “predictions indicate a gradual reduction in age-standardized rates.” Could the authors briefly state which statistical model was used for this projection in the abstract?

Author response #4: We appreciate the reviewer's valuable feedback. We agree that the statistical model used for the projection should be mentioned in the abstract for clarity. We have added this information to the abstract as suggested. We used the Nordpred model for our projections, which is an improved version of the Age-Period-Cohort (APC) regression framework specifically designed for predicting trends in health outcomes. This information was previously included in our Methods section but was missing from the abstract. We have now added this detail to ensure methodological transparency in the abstract.

Reviewer point #5: Method: It is mentioned that stroke burden below 25 years old was assumed negligible. Is there any literature supporting this cutoff, or was this assumption made based on data limitations?

Author response #5: We thank the reviewer for this insightful question. We have revised our methodology section to provide better justification for our age cutoff. Our decision to use 25 years as the lower age limit is supported by both epidemiological evidence and previous research methodologies. Stroke incidence is known to increase significantly with age, with the vast majority of cases occurring in individuals over 25 years. This approach aligns with methodologies used in similar GBD-based analyses of stroke burden, including Hu et al. (2024), Lu et al. (2021) and Jiang et al. (2020), which employed similar age stratification approaches. We have updated the Methods section to provide this clarification and added appropriate references.

Reviewer point #6: Results: The findings suggest a global decline in age-standardized rates, yet absolute numbers continue to rise. Could this paradox be further elaborated upon? For example, how do population growth and aging contribute to the observed trends in the Discussion section?

Author response #6: We thank the reviewer for their valuable comment regarding the need to clarify the paradox between declining age-standardized rates and rising absolute numbers of stroke burden attributable to household air pollution (HAP). In response to this suggestion, we have significantly refined the Discussion section to provide a more comprehensive explanation of this observation. Initially, we expanded the third paragraph of the Discussion to highlight the roles of population growth and aging as key drivers of the increasing absolute burden, despite reductions in age-standardized rates. Building on the reviewer's feedback, we have further enriched this discussion by addressing how the rising absolute numbers may reflect the limited effectiveness of current primary stroke and cardiovascular disease prevention strategies, particularly in low- and middle-income countries (LMICs), as well as disparities in stroke care delivery, accessibility, and workforce capacity in these regions. These systemic factors, combined with demographic trends, provide a more holistic explanation for the observed patterns. The revised paragraph (paragraph 3 of the Discussion section) is presented below. We are grateful for the reviewer's suggestion, which has significantly improved the depth and clarity of our analysis.

Reviewer point #7: Results: All figures are too small and blurred, this makes interpretation very difficult.

Author response #7: We sincerely thank the reviewer for pointing out the issue regarding the size and clarity of the figures. We acknowledge that the figures in the initial submission were not sufficiently clear, which may have hindered their interpretation. To address this concern, we have revised all figures to ensure they are of higher resolution and appropriately sized for better readability. We have increased the resolution to 300 dpi and adjusted the dimensions to make the text, labels, and data points more legible. We hope these improvements will enhance the clarity and usability of the figures in the revised manuscript. Thank you again for your valuable feedback, which has helped us improve the quality of our work.

Reviewer point #8: Discussion: The study finds that males experience a higher burden of HAP-induced stroke. However, it is unclear if this is due to increased exposure to air pollution or other co-existing risk factors (e.g., smoking, hypertension). Could this be discussed further?

Author response #8: We appreciate the reviewer’s insightful comment. We have expanded the discussion on potential drivers of the gender disparity in the revised manuscript (Discussion section, paragraph 4). Specifically, we now elaborate on the interplay between HAP exposure, occupational risks, and behavioral factors. We highlight that while women may experience prolonged indoor exposure due to traditional cooking roles, men’s higher prevalence of smoking, hypertension, and occupational exposures (e.g., in high-pollution industries) likely contribute to their elevated stroke burden. Citations to supporting literature have been added to strengthen these points.

Reviewer point #9: Discussion: Was occupational exposure considered as a potential confounder? In some developing regions, men may be more likely to work in high-exposure settings, which could affect results.

Author response #9: We thank the reviewer’s thoughtful comments and suggestions that have helped improve our manuscript. While the GBD 2021 dataset does not explicitly adjust for occupational exposure, we acknowledge this limitation and have added a discussion on its potential confounding effects (Discussion section, paragraph 4). We propose that occupational exposure in high-risk industries (e.g., mining, construction) may synergize with HAP to exacerbate stroke risk in males. We emphasize the need for future studies to integrate occupational exposure data for a more comprehensive risk assessment.

Reviewer point #10: Discussion: The study finds that middle-SDI regions have the highest burden. However, previous research often highlights low-SDI regions as suffering the most from air pollution exposure. Could the authors discuss why their findings diverge?

Author response #10: We appreciate the reviewer’s request to explain why our finding—that middle-SDI regions exhibit the highest burden of HAP-induced stroke—diverges from previous research, which frequently emphasizes low-SDI regions as the most affected by air pollution exposure. To address this, we have revised the fifth paragraph of the "Discussion" section (originally beginning with "Significant differences exist in the stroke burden related to HAP across different regions and countries") to include a detailed discussion of this divergence. The revised paragraph now explains that: (1) in low-SDI regions, limited healthcare access and underreporting may lead to an underestimation of the stroke burden in the GBD 2021 dataset, masking its full extent; (2) middle-SDI regions, such as parts of East Asia and South Asia, face a unique dual burden from persistent household use of solid fuels and increasing ambient pollution due to rapid urbanization and industrialization, amplifying HAP-related stroke risk; and (3) unlike high-SDI regions, middle-SDI regions have not yet fully transitioned to clean energy, while potentially benefiting from better health data reporting compared to low-SDI regions, which may result in higher recorded burdens. This explanation highlights the complex relationship between socioeconomic development, exposure patterns, and data availability, positioning middle-SDI regions as a critical transitional phase of heightened vulnerability.

Reviewer point #11: Discussion: Some countries (e.g., Vietnam, Mongolia) experienced an increase in stroke burden due to HAP, while others (e.g., Estonia, Norway) showed a sharp decline. What policy or environmental factors might explain these differences?

Author response #11: We appreciate the reviewer's insightful comment regarding the stark differences in HAP-related stroke burden trends across countries. As noted in our findings, countries like Vietnam and Mongolia experienced significant increases in stroke burden due to HAP, while Estonia and Norway showed sharp declines. These disparities indeed warrant further discussion of underlying policy and environmental factors. In response to this valuable suggestion, we have expanded our discussion to include a detailed analysis of potential explanatory factors for these country-level variations. Specifically, we have added a new paragraph discussing how differences in energy transition policies, economic development trajectories, regulatory frameworks, geographic and climate conditions, and healthcare system development may contribute to these observed trends.

2.Reviewer 1

Reviewer point #1: The authors present a valuable study on the epidemiology of stroke attributed to household air pollution. Their findings indicate decreasing mortality, DALY, and YLL trends but an increasing YLD trend after age adjustment. However, these results may be significantly influenced by evolving treatment modalities that reduce both stroke-related mortality and morbidity, representing an important confounding factor. This limitation should be explicitly discussed.

Author response #1: We thank this comment. We agree that evolving treatment modalities for stroke could represent an important confounding factor influencing the observed trends of decreasing age-standardized mortality, DALYs, and YLLs, alongside an increasing YLDs trend. To address this, we have revised the manuscript by explicitly discussing this limitation in the Discussion section (Section 4), specifically in the paragraph addressing study limitations (the penultimate paragraph of the section). We have added a detailed explanation highlighting how advances in medical interventions—such as improved acute stroke care (e.g., thrombolysis and endovascular therapy), better secondary prevention strategies (e.g., antihypertensive and statin therapies), and enhanced rehabilitation programs—may have reduced mortality and severe disability while increasing the prevalence of individuals living with milder, long-term disabilities. We also note that these effects might be more pronounced in high-SDI regions, aligning with the regional disparities observed in our findings. This addition strengthens the interpretation of our results by acknowledging the potential impact of treatment advancements as a confounding factor.

Reviewer point #2: Additionally, in Figure 4, the YLD trend does not align with other indices specifically in high-middle SDI countries, which warrants further explanation. The study may also benefit from discussing potential future trends based on geographic distribution, such as SDI categories, to provide a more comprehensive perspective.

Author response #2: We thank the reviewer for their insightful observation and valuable suggestion. Regarding the discrepancy in the YLD trend in high-middle SDI countries as depicted in Figure 4, we agree that this divergence from other indices—such as deaths, DALYs, and YLLs—merits further clarification. In high-middle SDI regions, the age-standardized YLD rate shows a less pronounced decline, or in some cases a slight increase, compared to the consistent downward trends obs

---

## [Decision Letter · Decision Letter 1]

Global Stroke Burden Attributable to Household Air Pollution: Insights from GBD 2021 and Projections to 2040

PONE-D-25-02181R1

Dear Dr. Zeng,

We’re pleased to inform you that your manuscript has been judged scientifically suitable for publication and will be formally accepted for publication once it meets all outstanding technical requirements.

Kind regards,

Amir Hossein Behnoush

Academic Editor

PLOS ONE

Additional Editor Comments (optional):

Reviewers' comments:

Reviewer's Responses to Questions

**Comments to the Author**

1. If the authors have adequately addressed your comments raised in a previous round of review and you feel that this manuscript is now acceptable for publication, you may indicate that here to bypass the “Comments to the Author” section, enter your conflict of interest statement in the “Confidential to Editor” section, and submit your "Accept" recommendation.

Reviewer #1: All comments have been addressed

2. Is the manuscript technically sound, and do the data support the conclusions?

Reviewer #1: Yes

3. Has the statistical analysis been performed appropriately and rigorously? 

Reviewer #1: Yes

4. Have the authors made all data underlying the findings in their manuscript fully available?

Reviewer #1: Yes

5. Is the manuscript presented in an intelligible fashion and written in standard English?

Reviewer #1: Yes

6. Review Comments to the Author

Reviewer #1: (No Response)

7. PLOS authors have the option to publish the peer review history of their article (what does this mean? ). If published, this will include your full peer review and any attached files.

**Do you want your identity to be public for this peer review?** For information about this choice, including consent withdrawal, please see our Privacy Policy .

Reviewer #1: **Yes: ** Mohammad Amin Dabbagh Ohadi

---

## [Editor Report · Acceptance letter]

PONE-D-25-02181R1

PLOS ONE

Dear Dr. Zeng,

I'm pleased to inform you that your manuscript has been deemed suitable for publication in PLOS ONE. Congratulations! Your manuscript is now being handed over to our production team.

Kind regards,

on behalf of

Dr. Amir Hossein Behnoush

Academic Editor

PLOS ONE